CellPress

## Commentary

# ClinGen API platform for classification of human genetic variants

Neethu Shah,[1,10] Tierra Farris,[1,10] Arturo Alejandro Zuniga,[1] Andrew R. Jackson,[1] Jessie Arce,[1] Kevin Riehle,[1] Christine G. Preston,[2] Mark E. Mandell,[2] Bryan Wulf,[2] Gloria Cheung,[2] Keyang Yu,[1] Deborah I. Ritter,[3] Dubravka Jevtic,[4] Miroslav Milinkov,[4] Novak Martinovic,[4] Nevena Vucinic,[4] Aleksandar Mihajlovic,[4] Alan F. Rubin,[5,6] Melissa S. Cline,[7] Marina Distefano,[8] Malachi Griffith,[9] Obi L. Griffith,[9] Matt W. Wright,[2] Teri E. Klein,[2] Sharon E. Plon,[1,3] and Aleksandar Milosavljevic[1,*]

[1]Department of Molecular and Human Genetics, Baylor College of Medicine, One Baylor Plaza, Houston, TX 77030, USA
[2]Department of Biomedical Data Science, Stanford University School of Medicine, Stanford, CA 94305, USA
[3]Department of Pediatrics, Baylor College of Medicine, Houston, TX 77030, USA
[4]Persida Inc., 1180 46th Street, Brooklyn, NY 11219, USA
[5]Bioinformatics Division, The Walter and Eliza Hall Institute of Medical Research, Parkville, VIC, Australia
[6]Department of Medical Biology, University of Melbourne, Parkville, VIC, Australia
[7]UC Santa Cruz Genomics Institute, University of California, Santa Cruz, Santa Cruz, CA 95064, USA
[8]Medical and Population Genetics Group, The Broad Institute of MIT and Harvard, Cambridge, MA 02142, USA
[9]Washington University School of Medicine, St. Louis, MO 63110, USA
[10]These authors contributed equally
*Correspondence: amilosav@bcm.edu

In this commentary, we describe how the Clinical Genome Resource's (ClinGen's) application programming interface-based microservices accelerate growth and dissemination of knowledge about human genetic variation. By exposing findable, accessible, interoperable, reusable, and AI-ready variant data, ClinGen lays a foundation for next-generation software applications, AI systems, and variant classification workflows.

## Background

Over the last decade, over 2,800 contributors to the Clinical Genome Resource (ClinGen) developed a clinical genomics knowledge base to support clinical genetic diagnosis, precision medicine, and research. This knowledge base spans four curation domains (gene validity, actionability, variant pathogenicity, and dosage sensitivity) with recently added work on somatic variants in cancer. The work of over 59 gene curation expert panels (GCEPs), 80 variant curation expert panels (VCEPs), and 60 working groups follows the same three general steps: (1) aggregate relevant external information about genes, diseases, clinical interventions, and genetic variants; (2) curate the information based on standardized guidelines or ClinGen frameworks; and (3) disseminate curated content for use in clinical genetics and precision medicine.[1] The ClinGen expert and evidence-based curation is supported by a multi-layered software infrastructure, which includes databases, tools, and interfaces to support data aggregation, curation, and dissemination of curated knowledge. Here, we focus on the application pro-gramming interface (API) platform consisting of API microservices that support curation and pathogenicity classification of variants. Within the ClinGen variant curation framework, these API microservices support the three-step workflow, illustrated in Figure 1A, that is integrated and accessible within ClinGen's variant curation interface (VCI).[1]

ClinGen's software and infrastructure for variant curation are developed by multiple teams across institutions and cloud platforms, creating an integration challenge. To support seamless and collaborative data exchange across applications, we established ClinGen Data Exchange, a distributed event streaming service based on the Apache Kafka messaging system.[2] As the consortium and its resources matured, clinical genetics and informatics communities increasingly sought flexible, fully featured API services with findable, accessible, interoperable, and reusable (FAIR) ClinGen-curated knowledge and supporting data. The demand was recently amplified by the needs of commercial and academic artificial intelligence (AI) agents and related systems, which rely on ClinGen as an authoritative source of computable knowledge and expert curated information about clinically relevant genetic variation. To meet this demand, we changed the software architecture from a monolithic (Figure 1B) to an "API-first" microservice design pattern[3] (Figure 1C).

We developed the following four groups of API-accessible microservices (summarized in Table 1) that are generically implemented to support variant data aggregation, curation, and dissemination: (1) ClinGen Allele Registry (CAR: reg.clinicalgenome.org), which enables the registration and standardized naming of small and copy-number variants, using established variant representation formats such as Human Genome Variation Society (HGVS) nomenclature and variant call format; (2) Linked Data Hub (LDH: ldh.clinicalgenome.org), which links, excerpts, standardizes, permalinks, updates, and versions variant-related data from external sources, providing an aggregated, transparent, auditable, and current view of evolving knowledge and supporting data; (3) Criteria Specification Registry (CSpec: cspec.clinicalgenome.org), which delivers

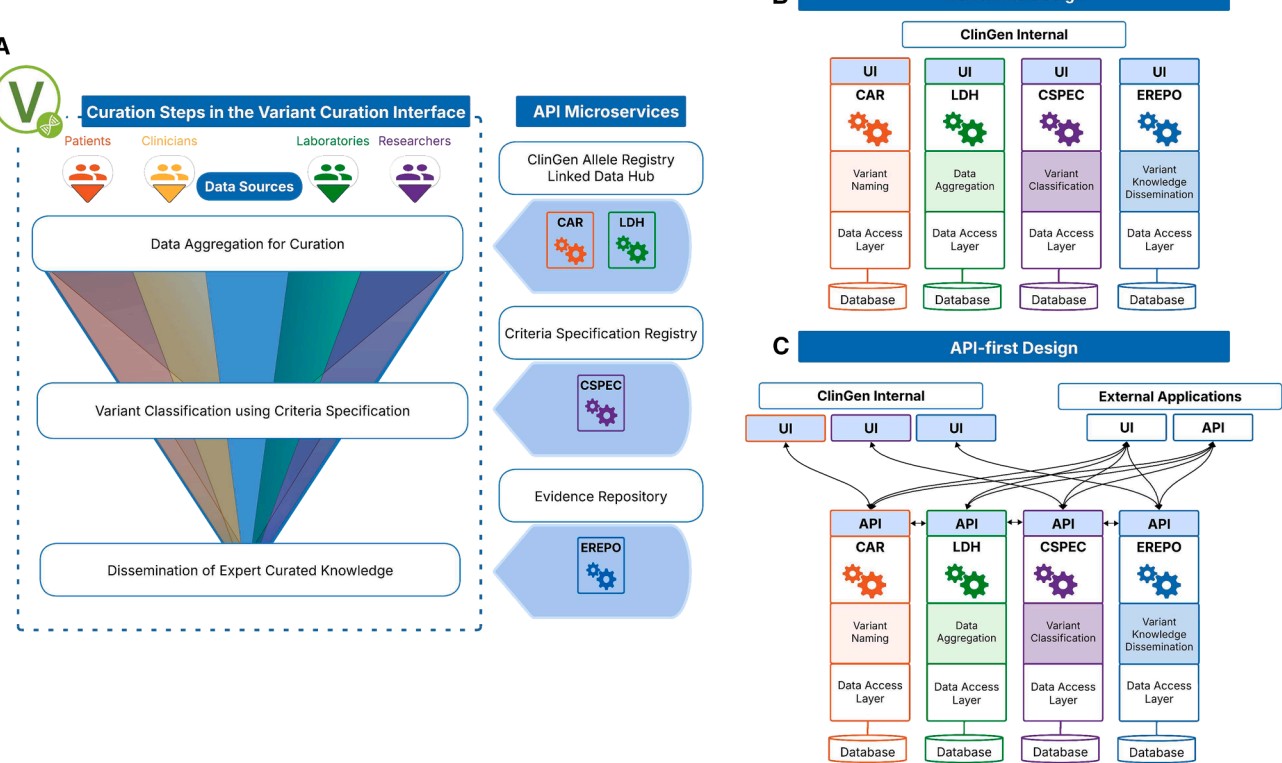

**Figure 1. ClinGen API microservices for classification of human genetic variants**
(A) Overview of API microservices originally designed to be accessed through ClinGens' variant curation interface (VCI). Data aggregation for curation: the CAR "registration" and "lookup" APIs name variants using globally unique identifiers, and the LDH APIs provide access to variant information. Variant classification using criteria specifications: the CSpec Registry API helps calculate variant pathogenicity by providing VCEP-defined assertion criteria according to current ACMG/AMP standards. Dissemination of expert curated knowledge: the ERepo API disseminates curated variant knowledge in versioned, structured, and standardized machine-readable formats.
(B and C) Monolithic vs. API-first design. Comparison of monolithic and API-first microservices illustrating use of API by both ClinGen internal applications such as the VCI and external applications and AI crawlers.

authoritative, structured, computable representations of gene- and disease-specific specifications aligned with American College of Medical Genetics and Genomics/Association for Molecular Pathology (ACMG/AMP) variant interpretation guidelines,[4] as developed by the ClinGen VCEPs; and (4) Evidence Repository (ERepo: erepo.clinicalgenome.org), which distributes variant classifications and supporting evidence in both human- and machine-readable formats to support transparent and interoperable sharing of variant knowledge.

### ClinGen Allele Registry
The CAR is a variant naming service that ensures unique and consistent naming of genetic variants across transcripts and genome builds. Its canonicalization algorithm follows in-memory sequence-alignment-based indexing to group variants

denoting the same nucleotide variant, insertion, or deletion and assigns them globally unique canonical allele identifiers (CAids).[1] The number of registered variants in the registry has increased by four orders of magnitude over the last decade. As of June 2025, the CAR has 2.96 billion registered variants, including single nucleotide variants (SNVs) and small indels. CAR extension API services support registration of additional variant types such as copy number variants (CNVs).

The CAR has a wide range of API endpoints that are optimized, multi-threaded, and highly efficient. For instance, the API for new variant(s) registration can assign identifiers at a rate of up to ~50,000 variants per second. One can also query and look up existing variants via CAid, HGVS nomenclature, additional identifiers from external databases, gene, reference genome, and position. As a

driver project for Global Alliance for Genomics and Health (GA4GH), the CAR has extended its APIs to support Variation Representation Specification (VRS), including dedicated endpoints for retrieving VRS allele objects using HGVS nomenclature and for resolving VRS computed identifiers from CAids.[5]

### Linked Data Hub
The LDH supports aggregation of information about variants by providing means to share variant information that is either accessible in external databases and literature or computed by algorithms. LDH stores links to variant information in external sources or excerpts of information in standardized formats, or both, that serve as supporting evidence for variant classification. LDH provides permalinks to timestamped excerpts to document supporting evidence at the time of variant

**Table 1. Applications with the corresponding API microservices and links to documentation**

| Application | API service endpoint and documentation | |
| --- | --- | --- |
| ClinGen Allele Registry https://reg.clinicalgenome.org/ | API: | https://reg.clinicalgenome.org/ |
| | DOC: | https://reg.clinicalgenome.org/docs/cg-car/ |
| Linked Data Hub https://ldh.clinicalgenome.org/ldh/srvc | API: | https://ldh.clinicalgenome.org/ldh/srvc |
| | DOC: | https://ldh.clinicalgenome.org/docs/ldh/api/ |
| Evidence Repository https://erepo.clinicalgenome.org/ | API: | https://erepo.clinicalgenome.org/evrepo/api/summary/srvc |
| | DOC: | https://erepo.clinicalgenome.org/docs/cg-erepo/ |
| Criteria Specification Registry https://cspec.clinicalgenome.org/ | API: | https://cspec.clinicalgenome.org/cspec/srvc |
| | DOC: | https://genboree.org/gitlab/clingen/cspec/cspec-registry/-/wikis/CSpec-Registry-API |

curation or classification. It supports versioning and update propagation that retains information about earlier records from the same source. The LDH model adapts the W3C Web Annotation Data Model, where subject (variant) and linked data entities (variant annotations) representing nodes in the graph model are linked to each other via specific edges. As of June 2025, over 909 million CAR variants are linked in the LDH to sources of evidence. Evidence types include allele frequency data (Genome Aggregation Database [gnomAD]), clinical significance classifications and evidence (BRCA Exchange, Clinical Interpretation of Variants in Cancer [CIViC], ERepo, ClinVar), and variant effect data and predictions (multiplexed assays of variant effect, rare exome variant ensemble learner, etc.)[6–8] For a full up-to-date list of included entities and predictors, see the LDH documentation at ldh.clinicalgenome.org/docs/ldh/overview.html#entity-types.

The LDH RESTful APIs facilitate efficient, programmatic access to collated links and select variant data in machine-readable JSON format for use by a variety of curation tools, including the VCI and the CAR user interface. Additionally, the flexible document model allows the LDH to support diverse entity types with sensible data models and accepted community data standards such as those from the GA4GH or W3C. The adoption of community data standards and standardized JSON format data allows interoperability with a variety of downstream tools.

## Criteria Specification Registry

The CSpec Registry is a public repository of authoritative, ClinGen VCEP-specified ACMG/AMP standards for genes and genetic disorders. The registry delivers information in human- and machine-readable formats, including JSON/JSON-LD via APIs and HTML via user interfaces. The CSpec Registry APIs allow tools like the VCI and other resources external to ClinGen to integrate the specifications into their variant classification and genome interpretation processes (see ClinGen-developed FAIR variant data and knowledge is widely accessed via API endpoints). As of June 2025, there are 108 publicly accessible specifications for more than 120 genes. One can query CSpec APIs to look up a specification document by gene, disease, or VCEP name. These APIs also allow queries to provide a summary of the assertion criteria codes across all published specifications.

## Evidence Repository

The ERepo is a distribution point for an FDA-recognized, public, human genetic variant database that contains structured, ClinGen-curated classifications of variant pathogenicity with supporting evidence. In addition to publishing variant classification records from the VCI, the ERepo API has comprehensive functionalities to search, filter, and download classifications. The supporting evidence and provenance for variant classifications are provided using the Scientific Evidence and Provenance Information Ontology (SEPIO) standard[9] and a variant interpretation data model, all accessible for the broader community via APIs in JSON and JSON-LD formats. As of June 2025, ERepo contains 10,527 curated variants for over 140 genes produced by 37 VCEPs. The ERepo versioning system provides version control for both historical and newly created classifications, making all versions publicly available via its APIs.

## ClinGen-developed FAIR variant data and knowledge are widely accessed via API endpoints

Over a twelve-month period, the CAR API service averaged 59 million requests per month, while ERepo and CSpec received approximately 415,000 and 140,000 monthly requests, respectively (summarized in Figure S1). By offering global persistent identifiers, standardized access protocols, support for community data standards, and a modular, scalable design, these APIs align with FAIR data principles. The APIs are utilized beyond ClinGen by a growing number of external clinical resource sites and software tools. Use of these microservices prevent duplication of effort while liberating other resources to focus on additional content and development of user interfaces that support specific use cases and cater to targeted audiences. For example, the Database of Genomic Variation and Phenotype in Humans using Ensembl Resources (DECIPHER)[6] integrates VCEP-specified variant classification criteria through the CSpec Registry APIs. These gene- and disease-specific ACMG criteria codes are cohesively incorporated into the DECIPHER sequence variant pathogenicity interface, supporting expert-curated variant interpretation. DECIPHER also exports VCEP-specific variant classifications through the ERepo download APIs. ClinVar has ERepo variant classification URLs and CAids integrated into its records, ensuring comprehensive and traceable variant content. The Shariant platform facilitates the sharing of variant interpretations and supporting evidence among Australian

clinical laboratories.[10] It extensively leverages the CAR APIs within its data processing pipelines for variant naming, normalization, and genomic coordinate lift-over. This enables accurate aggregation and comparison of variants across different genome builds (e.g., GRCh37 and GRCh38) and varying HGVS nomenclature representations. BRCA Exchange, a genomic resource that aggregates BRCA1 and BRCA2 data with over 20,000 expert classifications, utilizes the CAR and CSpec APIs to integrate CAids and VCEP-specified rules into its platform, respectively. Open-source genomic databases such as CIViC and gnomAD integrate CAR identifiers through APIs, promoting data consistency and interoperability.

### AI integration
Artificial intelligence (AI) and machine learning (ML) tools and agents are increasingly used to support variant classification. Our API usage tracking indicates that AI web crawlers account for approximately 2.4% of the API requests, suggesting active use of the ClinGen knowledge by AI systems. To address AI integration into scientific workflows, the NIH Bridge2AI program[11] proposed seven "AI-readiness" criteria for biomedical data, including data FAIRness. In addition to FAIRness, ClinGen API microservices address key AI-readiness recommendations identified by Bridge2AI, including provenance, characterization, pre-model explainability, ethics, sustainability, and computability, as summarized in Table S1.

ClinGen API platform extends the concept of FAIRness not only to data but also to computable knowledge in the form of assertions about variant pathogenicity with supporting evidence accessible via the ERepo. Unlike the data, the assertions about pathogenicity are general and derived through a curation process performed by the expert panels rather than being directly computed by a data processing pipeline. The assertions are backed by evidence according to standards of evidential reasoning documented in the CSpec Registry. The platform, therefore, supports the creation and consumption of explicit computable knowledge that has traditionally been shared in the form of scientific evidence-based arguments. Being structured and machine

readable, ClinGen knowledge is serializable in the form of knowledge graphs that are routinely used by modern AI frameworks such as graph-based retrieval-augmented generation (Graph RAG).[12]

Because much variant information relevant for curation and interpretation resides in literature, RAG is emerging as one of the most impactful AI frameworks in evidence curation and variant interpretation. The LDH is a gateway for RAG workflow integration, as it provides API access for both input and output of RAG tools with potential semi-autonomous AI agentic extensions[13] in the future. Regarding RAG inputs, one key challenge is the inconsistent naming of genetic variants in literature. To address this, in collaboration with the National Center for Biotechnology Information (NCBI), we indexed mentions of variants in literature using the CAR and developed a pipeline that stores in the LDH excerpts of variant mentions from PubMed Central open access articles in the W3C web annotation format.

As of June 2025, the LDH contains annotations for 48,457 PubMed open access articles indexed by tmVar3 and pubTator3.[14] These annotations contain 707,627 variant excerpts linked to 1,140,259 unique CAR variants, 21,707 genes, and 6,249 MONDO disease entities. By applying a standardized W3C annotation format to systematically identify, mark, and link variant mentions within PubMed articles to canonical variant and gene identifiers, unstructured textual information is transformed into a structured knowledge graph with clear provenance and context. This knowledge graph provides key input for Graph RAG applications, which in turn can store their results within LDH. As we enter the era of agentic AI, the APIs will increasingly provide access points for AI agents to perform autonomous or collaborative actions. The API microservices therefore provide a semantic layer for AI workflow integration and reasoning across previously disconnected sources of information to support variant interpretation, research hypothesis generation, and precision medicine.

### ClinGen Pathogenicity Calculator as a model application built on the APIs
The API microservices support variant classification tools like the ClinGen Pathogenicity Calculator, enabling user inter-

faces and software development both within and beyond ClinGen. The calculator caters to the needs of external groups who use ClinGen knowledge and data in combination with their own in-house resources. In contrast to ClinGen's VCI, which supports comprehensive evidence curation and classification of variants by ClinGen experts, the calculator automates the formal reasoning and provisional assertions of variant pathogenicity. The calculator utilizes CAR APIs to look up canonical variant identifiers and CSpec APIs to fill the evidence codes panel to classify variants as per the applicable combination of rules. By leveraging key functionality of ClinGen APIs, the calculator validates the utility of APIs as a resource to accelerate software application development outside of ClinGen.[15] The software is available under a permissive open-source license and can be used as a starting point for software development by entities outside of ClinGen.

### Conclusion and discussion
We have developed a comprehensive set of API microservices to support ClinGen's variant curation processes and demonstrated the utility of this platform for both variant curation within ClinGen using the VCI and external application development such as Pathogenicity Calculator, a model for external application development that utilizes the APIs. These microservices are used widely beyond the consortium to catalyze variant interpretation and classification by leveraging ClinGen knowledge, the latest ClinGen frameworks, and professional guidelines. With the advent of AI-generated software, we envision a future where even non-programmers may develop interfaces and software applications on top of this API platform.

As agentic AI solutions enter variant interpretation and curation workflows, the API microservices described here will be essential for their rapid adoption and evolution. They implement FAIR principles and other AI-readiness criteria by providing canonical variant identifiers, API-accessible variant information, structured variant classifications, and standards for determining variant pathogenicity. These services extend the concept of FAIRness not only to data but also to computable knowledge in the form of assertions about variant

pathogenicity that are backed by supporting evidence according to authoritative criteria of evidential reasoning. The computable knowledge about human genetic variation within the platform provides a semantic layer for AI reasoning across previously disconnected sources of information. Taken together, the API-centric microservices platform is poised to catalyze the generation and dissemination of variant data and knowledge by both ClinGen and the wide research and clinical community.

## ACKNOWLEDGMENTS

ClinGen is primarily funded by the National Human Genome Research Institute (NHGRI), with co-funding from the National Cancer Institute (NCI), through the following grants: Baylor/Stanford U24HG009649, Broad/Geisinger U24HG006834, and UNC/Kaiser U24HG009650. M.G. and O.L.G. were supported by NCI grants U24CA275783, NIH NCI U24CA258115, and NIH NCI U24CA237719. M.S.C. is supported by NCI grants 5R01CA264971 and 5U24CA258058. Additional partial funding for this work was provided by the NIH Common Fund under "other transaction" OT2 OD030547 to A. Milosavljevic and by the Henry and Emma Meyer Chair in Molecular Genetics to A. Milosavljevic. The content is solely the responsibility of the authors and does not necessarily represent the official views of the National Institutes of Health or other funders.

The authors thank Anton Kodochygov (Baylor College of Medicine) for addressing and improving select deficiencies in the ClinGen Allele Registry. The authors also thank Liam E. Mulhall (Stanford University) for his technical discussions with the ClinGen Baylor and Stanford VCI development teams.

## DECLARATION OF INTERESTS

S.E.P. is a member of the Baylor Genetics scientific advisory panel. A. Milosavljevic owns shares in IP Genesis, Inc., which provides bioinformatic consulting services.

## AUTHOR CONTRIBUTIONS

N.S.: conceptualization; writing – original draft; project administration; software. T.F.: conceptualization; writing – original draft; software. A.A.Z.: software. A.R.J.: software; supervision. J.A.: software. K.R.: project administration. C.G.P.: project administration. M.E.M.: software. B.W.: software. G.C.: software. K.Y.: project administration. D.I.R.: project administration; writing – review and editing. D.J.: software. N.M.: software. M.M.: software. N.V.: soft-

ware. A. Mihajlovic: project administration. A.F.R.: project administration; writing – review and editing. M.S.C.: project administration; writing – review and editing. M.D.: supervision; writing – review and editing. M.G.: supervision; writing – review and editing. O.L.G.: supervision. M.W.W.: supervision; writing – review and editing. T.E.K.: supervision; writing – review and editing. S.E.P.: supervision; writing – review and editing. A. Milosavljevic: conceptualization; supervision; writing – original draft.

## SUPPLEMENTAL INFORMATION

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
