## [Document S2. Transparent peer review records for Shah et al. · Cell Genomics]

Summary

Initial submission: Received : 9/4/2025

Scientific editor: Laura Zahn

First round of review: Number of reviewers: 2
Revision invited : 12/4/2025
Revision received : 1/9/2026

Second round of review: Number of reviewers: 1
Accepted : 3/8/2026

Data freely available: N/A

Code freely available: Yes

This transparent peer review record is not systematically proofread, type-set, or edited. Special characters, formatting, and equations may fail to render properly. Standard procedural text within the editor's letters has been deleted for the sake of brevity, but all official correspondence specific to the manuscript has been preserved.

Referees' reports, first round of review

Reviewer #1: ClinGen is an NIH-funded resource to evaluate genes, mutations and genetic variants for clinically relevant features and disorders. It is based on expert panels that curate genes for the clinical (genetic) and experimental evidence to be disease-causing. These curations will be permanently re-evaluated to ensure that the content on the website is overall up-to-date.

A major challenge are the huge amounts of data that have to be interchanged between various software systems. The article entitled "ClinGen API platform for classification of human genetic variants" by N. Shah & T. Farris and coworkers aims to describe the Application Programming Interface (API) platform. The manuscript is well written and informative.

I have indeed no comments to make on the manuscript; I appreciated reading it already in its current version.

Reviewer #2: This manuscript describes a set of APIs the ClinGen team has made available to provide open access to four resources which are of high value for human variant annotation/classification. It provides a useful summary of the APIs and a discussion of the overall design philosophy and example use cases. As such, it would be better described as a resource or technology article rather than a commentary. The report is well written and will be of great interest to bioinformaticians who are not aware of these services. It would be improved by a few minor changes.

The citations are very incomplete, listing mostly ClinGen papers and publications guiding the design approach. It would be more appropriate to cite other resources listed, for example ClinVar, MaveDB, DECIPHER, Shariant, gnomAD etc, by publication rather the website.

As the APIs are described as comprehensive, it would be useful to know are resources chosen for integration into the Linked Data hub. For example, why was only REVEL selected from the many prediction packages available?

There are minor issues with unnecessary capitalisation (e.g. 'User Interface' on page 8, 'Precision Medicine' on page 10) and some sentences which could be improved (e.g. 'Both human and machine-readable formats are provided via API (JSON and JSON-LD), and User Interface (HTML)').

Figure 1A suggests patients interact with the data aggregation interface; is this correct or does it represent a data type? If this is correct how this is managed to ensure patient support and data quality?

The list of compliance with Bridge2AI criteria listed in the supplementary information would benefit from clarification in places. The comment on compliance with reusability is that the APIs are publicly accessible, but accessibility is a different criterion and the reusability recommendation is 'Attach a clear and accessible data usage license that allows the responsible use of AI/ML'. Are licenses clearly available? The statement 'CSpec, ERepo and LDH contain complete data for a subset of ClinGen relevant genes and variants.' should be qualified to explain what is meant by 'complete'. There are well known biases in the collection of genomic data on which some of these resources depend, which needs to be made clear especially to the developers of AI tools who may be lacking knowledge of the area.

Authors' response to the first round of review

Reviewer #1: ClinGen is an NIH-funded resource to evaluate genes, mutations and genetic variants for clinically relevant features and disorders. It is based on expert panels that curate genes for the clinical (genetic) and experimental evidence to be disease-causing. These curations will be permanently reevaluated

to ensure that the content on the website is overall up-to-date.

A major challenge are the huge amounts of data that have to be interchanged between various software systems. The article entitled "ClinGen API platform for classification of human genetic variants" by N. Shah & T. Farris and coworkers aims to describe the Application Programming Interface (API) platform. The manuscript is well written and informative.

I have indeed no comments to make on the manuscript; I appreciated reading it already in its current version.

Reviewer #2: This manuscript describes a set of APIs the ClinGen team has made available to provide open access to four resources which are of high value for human variant annotation/classification. It provides a useful summary of the APIs and a discussion of the overall design philosophy and example use cases. As such, it would be better described as a resource or technology article rather than a commentary.

The report is well written and will be of great interest to bioinformaticians who are not aware of these services. It would be improved by a few minor changes.

1. The citations are very incomplete, listing mostly ClinGen papers and publications guiding the design approach. It would be more appropriate to cite other resources listed, for example ClinVar, MaveDB, DECIPHER, Shariant, gnomAD etc, by publication rather the website.

This is indeed a good point and thank you for pointing this out. Cell Genomics Commentary format permits a maximum of 15 references and hence most of the external databases were "cited" through the corresponding websites. We have remedied this by doing the following:

a) ClinGen tools and resources - Allele Registry, Variant Curation Interface now cite a common publication, the ClinGen marker paper (The Clinical Genome Resource (ClinGen): Advancing genomic knowledge through global curation. Genetics in Medicine, 2025.)

b) All the external resources listed are now cited directly by the corresponding publication or by common review articles to get the appropriate credit.

2. As the APIs are described as comprehensive, it would be useful to know are resources chosen for integration into the Linked Data hub. For example, why was only REVEL selected from the many prediction packages available?

Thank you for this question. External data entities in the ClinGen Linked Data Hub (LDH) are incorporated based on several factors, with stakeholder (invested user) demand being a primary driver. For example, support for REVEL scores was integrated into the LDH as a result of our direct collaboration with the ClinGen Stanford Variant Curation Interface (VCI) team. This collaboration ensures that both the REVEL scores and their associated transcript information are accessible within the VCI via the LDH API. However, as a part of our ongoing work links and

Response to Reviewers

excerpts from other algorithmic prediction tools are also available in the LDH. Also find below a list of all the in silico prediction tools that are in the LDH now. Please note that the manuscript mentions and refers to an exhaustive list of all the LDH entities here -

<https://ldh.genome.network/docs/ldh/overview.html#entity-types>.

LDH Collection	Predictors
OpenCRAVAT	Alphamissense, BayesDel, CADD Exome, DANN, ESM1b, FATHMM, FATHMM MKL, FATHMM XF, GERP++, MetaLR, Metarnn, MetaSVM, mistic, Mutation Assessor, MutationTaster, MutPred-v1, PhDSNPg, PhyloP, PrimateAI, PROVEAN, SIFT, VARIETY_R, VEST
RevelScore	REVEL
InSilicoPredictionScoreStat ementEmbedded	S-CAP 3 prime core dominant, S-CAP 3 prime core recessive, S-CAP 3 prime intronic, S-CAP exonic, S-CAP 5 prime core dominant, S-CAP 5 prime core recessive, S-CAP 5 prime extended, S-CAP 5 prime intronic

3. There are minor issues with unnecessary capitalisation (e.g. 'User Interface' on page 8, 'Precision Medicine' on page 10) and some sentences which could be improved (e.g. 'Both human and machine-readable formats are provided via API (JSON and JSON-LD), and User Interface (HTML)').

The manuscript has been carefully edited to remove unnecessary capitalisation and improve sentence structure.

4. Figure 1A suggests patients interact with the data aggregation interface; is this correct or does it represent a data type? If this is correct how this is managed to ensure patient support and data quality?

The ClinGen Linked Data Hub, the Variant Curation Interface, and the CSpec Registry do not engage directly with patients. Instead, these systems are used by domain experts and expert panel coordinators during various stages of variant curation and criteria specification. In Figure 1A, the label "Patients, Clinicians, Researchers, Laboratories" is intended to represent sources of data rather than direct users of the interfaces. To avoid confusion, the figure has been revised to explicitly label these as data sources.

5. The list of compliance with Bridge2AI criteria listed in the supplementary information would benefit from clarification in places. The comment on compliance with reusability is that the APIs are publicly accessible, but accessibility is a different criterion and the reusability recommendation is 'Attach a clear and accessible data usage license that allows the responsible use of AI/ML'. Are licenses clearly available? The statement 'CSpec, ERepo and LDH contain complete data for a subset of ClinGen relevant genes and variants.' should be qualified to explain what is meant by 'complete'. There are well known biases in the collection of genomic data on which some of these resources depend, which needs to be made clear especially to the developers of AI tools who may be lacking knowledge of the area. The statement 'CSpec, ERepo and LDH contain complete data for a subset of ClinGen relevant genes and variants.' should be qualified to explain what is meant by 'complete'

Thank you for pointing this out. The statement in Table 2 has been updated to specifically mention sources of bias that may impact these datasets. The sentence has been changed to - "CSpec, ERepo and LDH contain variant curation data for a subset of ClinGen relevant genes and variants. ClinGen primarily focuses on mendelian genetic disorders and therefore clinically relevant genes and their variants in this category are likely over-represented compared to other genes and variants in these datasets. Additionally, genomics data historically over-represents individuals of European ancestry and that bias may be present in these datasets as well."

Please note the revised comment in Table 2 about reusability license and data terms of use as given below.

"Data usage terms in general are mentioned in the ClinGen website -

<https://www.clinicalgenome.org/docs/terms-of-use/>. Data usage terms are also accessible through specific API endpoints as well. For example, note the LDH API endpoints for both the software license (<https://ldh.genome.network/ldh/srvc/license>) and the data terms of use

(<https://ldh.genome.network/ldh/srvc/license/tou>)."

Referees' reports, second round of review

Reviewer #2: I thank the authors for their clarifications.

Authors' response to the second round of review

none